# To assess the determinants of family planning uptake among women of reproductive age in rural settings, Morogoro Region, Tanzania. Protocol for a cross-sectional study

**Margareth Danielistan Machange** [1] *, **Mariam John Munyogwa** [2]

**1** Department of Clinical Nursing, School of Nursing and Public Health, University of Dodoma, Dodoma, Tanzania, **2** Department of Community Medicine, School of Medicine and Dentistry, University of Dodoma, Dodoma, Tanzania

* machanmargareth@gmail.com

**Funding:** The authors received no specific funding.

**Competing interests:** The author declared that no competing interest exist.

## Abstract

### Background

Maternal health outcomes in Tanzania had continued to be of great concern. According the Tanzania Demographic and Health surveys, maternal mortality is still unacceptable higher. Effective use of family planning is reported to prevent maternal deaths by more than 30%. However, the prevalence of family planning uptake is still lower especially in rural areas of this country despite the fact that family planning services are provided freely and at a subsidized cost in all public and private health care facilities respectively.

### Objective of the study

The broad objective is to assess the determinants of family planning uptake among women of reproductive age in rural settings, Morogoro Region, Tanzania.

### Methods

This study will be a cross-sectional study that will be conducted in rural areas of Morogoro Region. Study population will be women of reproductive age. Qualitative and quantitative approaches will be used to collect the data. Sampling techniques will involve stratification of urban and rural wards, simple and systematic random sampling for selection wards and households respectively and kish selection table for the selection one participant from a household. Purposive sampling will be applied to get participants for qualitative data. Data collection will be done by using semi-structured questionnaire and interview guide. Frequencies, percentages, chi-square test and logistic regression will be used to analyze the quantitative data whereas codes and themes formation will be used to analyze the qualitative data.

## Introduction

Maternal health outcomes in Tanzania had continued to be of great concern. According the Tanzania Demographic and Health survey (TDHS), there are 556 maternal deaths per 100,000 live births, which is unacceptable higher [1]. The family planning uptake information and services are fundamental to the health and human rights of all individuals [2]. Effective use of family planning is reported to prevent maternal deaths by more than 30% [3]. In Tanzania, the uptake of family is still low despite that there is slight steady increase. For instance, data from three consecutive Tanzania and Demographic Health surveys reports; 2004–05, 2010 and 2015–16 shows that the uptake of family planning had remained low 20%, 27% and 32% respectively [1]. This is only half way based on the national target of 60% of family planning uptake by 2020 for women in need [4]. The uptake of family planning is much lower in rural areas compared to urban areas [5]. According to 2017 estimates, about 214 million women of reproductive age in developing regions have an unmet need for contraception [2]. Few previous studies done in sub-Saharan Africa have demonstrated several factors associated with family planning uptake including; marriage, limited access to contraception, long distances to the health facility, a limited choice of methods, a fear or experience of side-effects, cultural or religious opposition, poor quality of available services and gender-based barriers [2, 6–8].

The government of Tanzania, had put various strategies to ensure that individuals have an access to the contraceptive information and services. Such strategies include; Integrating family planning services as a component of the reproductive, maternal, newborn, child, and adolescent health interventions provided by the Ministry of Health, Community Development, Gender, Elderly, and Children (MoHCDGEC) [9]. Furthermore, in the Health Sector Strategic Plan (HSSP-IV), the government is committed in ensuring family planning commodities are available and freely accessible to all women of reproductive age at all public health care facilities [9]. In addition, family planning services are also offered in private facilities at a subsidized cost [10]. However, despite the government efforts of ensuring adequate supply of FP commodities to all public health facilities at free cost to all medically eligible women, and deployment of skilled healthcare providers [10], the flow of FP services uptake is still low and particularly in rural settings [7, 11].

Morogoro is among 31 regions of Tanzania. The region has a total of seven districts which include one municipal council (Morogoro Municipal Council) and six district councils (Kilosa District Council, Morogoro District Ccouncil, Kilombero District Council, Ulanga District Council, Mvomero District Council and Gairo District Council. The six district councils are predominantly rural [12]. More than 71% of its population are farmers located in rural settings [13]. According to the information from the office of the Medical Officer In charge, Morogoro Rural has a total of 391 health facilities (15 hospitals, 42 health centers and 334 dispensaries). All these health care facilities provide family planning services. Maternal mortality rate (415/ 100,000) per live birth, fertility rate (4.6) and average number of children per woman (5.1) [1] are considerably high in the Region. Morogoro is among the regions with high proportion (94%) of women of reproductive age (15–49 years) with an exposure to sexual intercourse and high prevalence (38.5%) of teenagers who have started childbearing [1]. Further, the study conducted in rural setting of Morogoro Region reveal that the fertility rate and average number of children per woman were 6.1 and 8.0 which are higher than that of the Region [14] and the prevalence of family planning uptake is still low [15]. Therefore, the broad objective of this study is to assess the determinants of family planning uptake among women of reproductive age in rural areas of Morogoro Region. To achieve this broad objective, the specific objectives are:

1. To determine the prevalence of family planning uptake among women of reproductive age in rural areas of Morogoro Region.

2. To determine the socio-cultural factors associated with family planning uptake among women of reproductive age in rural area of Morogoro Region.

3. To determine the maternal related factors associated with family planning uptake among women of reproductive age in rural area of Morogoro Region.

4. To explore challenges, barriers and opportunities of family planning uptake among women of reproductive age in rural areas, Morogoro Region.

The results from this study will help programme planners and policy makers to plan for effective intervention strategies for increasing awareness and uptake on family planning among women of reproductive age. It is expected that the findings of this study will help towards reducing infant and maternal death, unwanted pregnancy and teenage pregnancy for sustainable development goals 3.1 and 3.2. The findings of this study will also serve as the base-line information for the future scholars regarding family planning in Tanzania and other similar settings worldwide.

## Methods

### Study area

Morogoro Region has total number of seven districts namely; Morogoro Municipal Council, Morogoro District Council, Kilosa District Council, Kilombero District Council, Mvomero District Council, Ulanga District Council and Gairo District Council. The Region has a total population of 2,218,492 and an annual growth rate of 2.4% [12]. The main ethnic groups are Wandamba, Wapogoro, Wasagara, Waluguru, Wakaguru. The Region is geographically bordered by the Tanga Region to the North, Coastal Region to the East, Ruvuma Region to the South and Iringa and Dodoma Regions to the West [16].

The Region maternal mortality rate is 415/100,000 per live birth, fertility rate is 4.6 and average number of children per woman is 5.1. About 94% of women (15–49 years) had an exposure to sexual intercourse, 38.5% of teenage have started childbearing [1]. The region is among the top 5 regions with highest prevalence of adolescent pregnancies (39%) [1, 16] in the country. The overall prevalence of family planning uptake in Morogoro Region is low (32%) [1].

More than 71% of the region is rural settings comprising farmers and livestock keepers [13]. Morogoro Rural has a total of 391 health facilities (15 hospitals, 42 health centers and 334 dispensaries). All these health care facilities provide family planning services.

### Research design

The study design will be a cross-sectional. The study will use both quantitative and qualitative approaches to assess the determinants of family planning uptake and among women of reproductive age in rural areas, Morogoro Region.

### Study population

The population for this study will be women of reproductive age from 15 to 49 years. The followings will be excluded from this study; pregnant women, women with less than three months in the study area and women with mental illness.

## Sample size calculation and sampling techniques

**Quantitative data.** The minimum sample size (N) for this study will be estimated by using the formula $n = z^2 p \frac{1-p}{e^2}$ whereby: n = minimum sample size, z = 1.96, e = 5% and p = 28% (prevalence of family planning use in Morogoro District Council [16].

$n = 1.96^2 28 \frac{1-28}{5^2} = 310$. The calculated sample size will be further adjusted for 10% attrition and the final calculated sample size will be n = 344.4. Therefore, an estimated total of 345 will be enrolled for this study.

Stratified sampling technique will be used to get rural and urban predominant districts. Simple random sampling by using rotary method will be used to select two rural districts. Six wards will be selected by using Microsoft excel random number generator. Households for the study will be selected by systematic random sampling technique whereby for each day of the study, first household will be selected by simple random sampling and thereafter every third household was selected until the end of the day. Kish selection table will be used to select one participant from the list of eligible participants in each selected household.

**Qualitative data.** The number of participants for qualitative part of this study will be determined by the saturation level of the required information from the participants. A purposive sampling will be employed to select key informants to participate in this study. Those key informants will be identified during the interview for quantitative data. The criteria will include the number of years on family planning uptake whereby at least those with more than three months on family planning will be included. Another criteria will be those who do not currently use family planning because of various reasons and those who seem to be familiar with many issues regarding family planning based on interview.

## Data collection tools and methods

**Quantitative data.** Data will be collected by using a developed semi-structured questionnaire (see S1 Appendix). The questionnaire has been adapted from TDHS [1] and modified to suit this study. The questionnaire has been organized into four parts; Part 1: Demographic and obstetric information, Part II: Family planning uptake, Part III: Cultural factors on family planning uptake, and Part IV: Challenges on family planning uptake. Participant's information will be obtained through face to face interview. Data collection will be done by trained research assistants who will be a nurse practitioner. Prior to interview, researchers/research assistants will establish the rapport and explain the purpose of the study to the study participants. Participants will be assured of confidentiality and freedom of participation in the study. Written or verbal informed consent will be sought from participants before the interview. The interview will take the duration of 10 to 20 minutes.

**Qualitative data.** In-depth interview will be conducted to collect the information. An interview guide (S2 Appendix) has been developed to guide the interviewer during the interview. The guide consists of 16 open-ended questions. The questions have been structured to capture the information regarding the challenges, barriers and opportunities for women to utilize family planning services. The tool has been developed inorder to expand the information from the participants after the quantitative interview. Informed consent (both verbal and written) will be sought from the participant before the interview. To ensure confidentiality of the information the following will be done; Firstly, interview will be conducted at a comfortable place where there is maximum privacy. Secondly, participant's quote will be referred by the pseudo names that will be assigned by the researcher. The actual names will not be used. The in-depth interview will be conducted by the principal researcher for uniformity and inorder to minimize biases. The interview will be recorded by special tape recorder and it is expected to last for 45–60 minutes. The informant will be interviewed only once.

**Definition and measurement of variables.**

i. **Dependent variable (Family planning uptake)** The dependent variable for this study will be family planning uptake. Family planning uptake will be defined as the participant who will report to use any type of contraceptive method within the current three months and be able to mention it.

ii. **Independent variables** The independent variables for this study will be: demographic and obstetrics characteristics, utilization of ANC and postnatal services, socio-cultural factors, challenges, barriers and opportunities of family planning uptake.

## Data analysis

**Quantitative data.** Data collected will be coded, cleaned and transformed by using the SPSS version 26 for WINDOWS computer program (SPSS Inc. Chicago). Descriptive analysis will be carried out to present frequency distributions for demographic and obstetric characteristics and family planning practices of the study participants. Chi-square test for independence will be conducted to compare prevalence of family planning uptake according to participants' selected characteristics. Thereafter, univariable and multivariable logistic regressions will be conducted to determine the associates of family planning uptake. All independent variables with p-value less than or equal to 0.25 at binary logistic regression models will be included in the multivariable logistic regression model. All probabilities will be two-tailed and independent variables with p-values $< 0.05$ will regarded as significant.

**Qualitative data.** Data will be analyzed by thematic analysis. NVivo software may be used to facilitate qualitative data analysis. Data will be closely examined to identify common themes- topic, ideas and patterns of meaning that come up repeatedly. Analysis will follow the following steps; Familiarization, coding, generating themes, reviewing themes, defining and naming

i. *Familiarization*. The researcher gets to know the data. This was done through transcribing audio reading through the text and taking initial notes, and generally looking through the data to get familiar with it.

ii. *Coding*. Researcher generate codes from the data. Coding will be done through highlighting sections of text usually phrases or sentences and coming up with shorthand labels or "codes" to describe their content.

iii. *Generating themes*. Themes are generally broader than codes. The researcher combined several codes into a single theme. The researcher looked over the codes created, identify patterns among them, and start coming up with few themes.

iv. *Reviewing themes*. Researcher will make sure that themes generated are useful and accurate representations of the data. This will be done through returning to the data and compare themes against data. Looking if themes are really present in the data or if there is anything to change to make themes work better.

v. *Defining and naming themes*. After having a final list of themes, the researcher will name and define each of them.
Defining themes involves formulating exactly what will mean by each theme and figuring out how it helps in understanding the data.

vi. *Writing up*. Finally, the researcher will write up the report of the analyzed data.

### Dissemination of results

The findings of this study will be presented at the followings; University of Dodoma, Morogoro Medical officer in-Charge office, Ministry of Health, Community Development, Gender, Elderly and Children. Furthermore, manuscript will be prepared and submitted at a peer reviewed journal for publication and presented at local and international conference.

### Ethical clearance and consent to participate

This study was submitted to the Directorate of Research, Publications and Consultancy of the University of Dodoma for ethical approval. The ethical committee has assessed and given the ethical approval for this study Ref. No. MA.84/261/02/185. Furthermore, the permission to conduct this study will be sought from the office of Morogoro Regional Administrative Secretary, District Medical Officer and ward respectively.

The participants will have the absolute right and freedom to withdraw from the study at any time with no effect to them. Confidentiality and anonymity will be maintained by use of code numbers on the questionnaire rather than names. Women who will be in need of family planning and those with related health challenges will be referred to the nearby health center for consultation and follow-ups.

## Discussion

The prevalence of family planning uptake in Tanzania is still low (32%) despite the government strategies to ensure at least 60% of people in need uses family planning by 2020 [4]. According the WHO reports, there are various factors which may lead to low family planning uptake particularly in developing world. Such factors includes; limited access to contraception, a limited choice of methods, a fear or experience of side-effects, cultural or religious opposition, poor quality of available services and gender-based barriers [2]. However, these factors may not be the same in all settings. Inorder to plan for cost effective interventions one need to understand the local context factors. For instance, previous scholars from different countries in Africa have shown that, among the factors that contributes to low family planning uptakes includes, poverty, low education, less exposure to media, rural residence, antenatal visits and delivery at the health care facility [5–8]. In Tanzania, there are limited information regarding the reasons for low family planning uptake in the country. Therefore, this study is aiming to assess the determinants associated with family planning uptake particularly in rural settings of the country. Morogoro Region is among the regions with higher maternal mortality rate 415/100,000 per live birth, fertility rate 4.6% and large family size (5.119) [1, 3, 12]. Further, the region has high proportion (94%) of women (15–49 years) reported to have had an exposure to sexual intercourse and high rates (38.5%) of teenage who have started childbearing [1]. Therefore, conducting this study in this region is important inorder to plan appropriate interventions to improve this situation.

This study is designed to be conducted in rural settings because of the following reasons; In Tanzania rural population comprised more than 60% of the total population. For Morogoro Region where this study will be conducted, the rural population comprises more than 71% of the total population in the region [13]. Previous studies had shown that family planning uptake is much lower in rural than urban settings [1, 2, 11]. Further, the prevalence of teenage childbearing and family size are relatively higher in rural settings. Furthermore, a study that was conducted in Morogoro showed that, the family planning uptake in the rural part of the region is very low (28%) [16]. Therefore, this study will help to inform the stakeholders about the factors that hinder family planning uptake in rural settings of Morogoro Region and also other similar settings in the country and elsewhere globally.

This study will use both quantitative and qualitative approaches (mixed approaches) to collect the data. The use of mixed approaches will help collection of many information which will adequately inform the stakeholders about the family planning uptake in the study area. Previous studies on family planning uptake in Tanzania have used quantitative approach [11, 16]. The use of qualitative approach will help to generate more factors that may not be adequately generated through the quantitative data.

The study will be conducted among women of reproductive age 15–49 years. This age group regardless of their status will give an opportunity of exploring determinants of family planning uptake to the more diverse groups of women including students, married and no married, nulliparaous and multipara etc. This is the most group that suffers the consequences of low utilization of family planning because it is biologically the sexually active and child bearing age. This group comprise of adolescent girls and young women who may not have completed their education and those who are still struggling to achieve their life goals. The prevalence of teenage childbearing (27%) [1] and the rate of abortion [17] are still unacceptable high in our country. In Morogoro Region where this study is expected to be conducted, it is among the regions with high prevalence of teenage child bearing (39%) [1]. Further, the region has high prevalence (94%) of women (15–49 years) who had an exposure to sexual intercourse [1]. Effective use of family planning is reported to reduce the rates of unintended pregnancies hence reduces HIV infection transmissions, unsafe abortion and improve maternal health [2]. This will benefit this group through enabling girls to complete their education and create opportunities for women to participate more fully in society, including paid employment [2]. Therefore, understanding the determinants of family planning uptake in this wider group will help the stakeholders to come up with the appropriate interventions.

## Limitations

The followings are the limitations to this study;

1. This study will be a cross sectional study hence not useful at establishing causal relationship between family planning uptake and their predictors.

2. All assessments will depend on self-reports by the participants with likely cause biases in some of the information.

3. The study participants will be females only. This will bias the information as males are also important group in family planning issues

## Supporting information

**S1 Appendix. Questionnaire.**
(DOCX)

**S2 Appendix. In-depth interview guide.**
(DOCX)

## Author Contributions

**Conceptualization:** Margareth Danielistan Machange, Mariam John Munyogwa.

**Methodology:** Margareth Danielistan Machange, Mariam John Munyogwa.

**Supervision:** Mariam John Munyogwa.

**Writing – original draft:** Margareth Danielistan Machange.

**Writing – review & editing:** Mariam John Munyogwa.

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
