## [Decision Letter · Decision Letter 0]

20 Oct 2021

PONE-D-21-12824

ASSESSING DETERMINANTS OF FAMILY PLANNING USE AMONG WOMEN OF REPRODUCTIVE AGE IN RURAL AREAS, MOROGORO, TANZANIA. MIXED CROSS-SECTIONAL  PROTOCOL STUDY.

PLOS ONE

Dear Dr. machange,

Thank you for submitting your manuscript to PLOS ONE. After careful consideration, we feel that it has merit but does not fully meet PLOS ONE’s publication criteria as it currently stands. Therefore, we invite you to submit a revised version of the manuscript that addresses the points raised during the review process.

We look forward to receiving your revised manuscript.

Kind regards,

Paola Viganò

Academic Editor

PLOS ONE

Additional Editor Comments (if provided):

The Authors need to address the issues raised by the two Reviewers. More details on the study design, sampling procedure and data collection are required. Moreover, limitations should be described.

Journal Requirements:

2. Please include additional information regarding the survey or questionnaire used in the study and ensure that you have provided sufficient details that others could replicate the analyses. For instance, if you developed a questionnaire as part of this study and it is not under a copyright more restrictive than CC-BY, please include a copy, in both the original language and English, as Supporting Information. If the original language is written in non-Latin characters, for example Amharic, Chinese, or Korean, please use a file format that ensures these characters are visible.

3. Please state whether you validated the questionnaire. Please provide details regarding the validation group within the methods section.

4. Please include a copy of the interview guide to be used in the study, in both the original language and English, as Supporting Information, or include a citation if it has been published previously.

7. Your ethics statement should only appear in the Methods section of your manuscript. If your ethics statement is written in any section besides the Methods, please delete it from any other section. 

Reviewers' comments:

Reviewer's Responses to Questions

**Comments to the Author**

1. Does the manuscript provide a valid rationale for the proposed study, with clearly identified and justified research questions?

Reviewer #1: Yes

Reviewer #2: Partly

2. Is the protocol technically sound and planned in a manner that will lead to a meaningful outcome and allow testing the stated hypotheses?

Reviewer #1: Yes

Reviewer #2: Partly

3. Is the methodology feasible and described in sufficient detail to allow the work to be replicable?

Reviewer #1: Yes

Reviewer #2: No

4. Have the authors described where all data underlying the findings will be made available when the study is complete?

Reviewer #1: Yes

Reviewer #2: Yes

5. Is the manuscript presented in an intelligible fashion and written in standard English?

Reviewer #1: Yes

Reviewer #2: Yes

6. Review Comments to the Author

You may also provide optional suggestions and comments to authors that they might find helpful in planning their study.

Reviewer #1: Comments to the authors

I enjoyed reading the study protocol, and it is an important topic and well presented. This protocol manuscript is also well written. However, I do have some points that require the authors' attention.

The discussion section must be added. This section should include the any issues involved in performing the study that are not covered in other sections. These can include: limitations of the study design & dissemination plans.

Sincerely,

Tesfalem Tilahun

Reviewer #2: The reviewer believes additional work would improve the quality of the manuscript before its publication.

ABSTRACT: Methodology

Authors indicate Objectives 1 through 3. However, readers do not know which objective you refer to 1, 2, or 3. Adding these numbers in the "Objective of the study" is recommended. Also, there is only one objective in the main text. Authors may consider including three specific objectives in the main text.

Keywords: "Women of Reproductive Age" would be more suitable than "Reproductive age."

Citation: Please be thorough on the citation format. Sometimes authors use (author, reference number), which is not consistent.

Introduction:

Women of Reproductive Age (WRA) appears first on this page. Please add an acronym where it first appears and be consistent in using it in subsequent text.

Authors define WRA are those aged 18-49. WHO and other UN agencies use the definition of WRA (15-49 years). Please provide annotation or explanation.

TDHS: where it first appears on page 4; please spell it out.

CycleBeads® is one word.

Literature review: it is suggested that authors add a review of more literature on contraceptive use among WRA in Tanzania (esp. Morogoro or neighboring regions) as many studies have already been conducted. By doing this, the authors will have a much more vigorous justification for conducting this study (including highlighting why this study has to be conducted among WRA in Morogoro and those residing in rural areas).

METHODS

Study Area: More description on Morogoro Region (the number of Districts within the Region, among which how many Districts are considered "rural," and how authors define "rural" in this study.

Research design: Do authors consider this study as a mixed-methods study? If so, please describe which design of a mixed-methods study the authors employ with its justification.

Sampling Procedure for Qualitative Approach

Please describe in more detail which approach of purposive sampling the authors adopt for this study.

How do authors determine key informants from the results of the quantitative study?

Data Collection Methods for Quantitative Approach

What kind of questionnaire do the authors plan to use for this study? Has it been validated?

Data Collection Methods for Qualitative Data

No details are known, but the authors' interview method in this manuscript would be key informant interviews rather than in-depth interviews.

Please also indicate whether recording and transcriptions are made, how long does each interview would take, and if the authors plan to interview the informants only once or iteratively.

Definition and Measurement of Variables

Would the dependent variables any FP use, rather than modern methods use?

During the qualitative interviews, authors should highlight on positive aspects of FP use rather than just focusing on the challenges.

7. PLOS authors have the option to publish the peer review history of their article (what does this mean?). If published, this will include your full peer review and any attached files.

Reviewer #1: **Yes: **Tesfalem Tilahun Yemane

Reviewer #2: No

---

## [Author Response · Author response to Decision Letter 0]

27 Dec 2021

POINT-BY-POINT RESPONSES TO REVIEWERS’ COMMENTS

We would like to take this opportunity to thank you all, editors and reviewers from PLOS ONE We have appreciated all the comments given in order to improve the manuscripts. We have thoroughly responded to all reviewers’ comments. The table below is a summary of responses for the reviewers.

 Comments to the authors Authors’ Response

 Authors had revised the whole document for grammar check and clarity

1 Please include additional information regarding the survey or questionnaire used in the study and ensure that you have provided sufficient details that others could replicate the analyses. For instance, if you developed a questionnaire as part of this study and it is not under a copyright more restrictive than CC-BY, please include a copy, in both the original language and English, as Supporting Information. If the original language is written in non-Latin characters, for example Amharic, Chinese, or Korean, please use a file format that ensures these characters are visible. The questionnaire description has been revised. The questionnaire has been adapted from the Demographic Health Surveys questionnaire. We only selected the questions related to family planning for this particular study. Minor modification has been done. 

The questionnaire has been uploaded as additional file (Appendix 1) (Page 6)

2 Please state whether you validated the questionnaire. Please provide details regarding the validation group within the methods section. The questionnaire was not validated because it was extracted from a standard questionnaire that is normally used for country wide survey. 

However minor modifications has been made 

3 Please include a copy of the interview guide to be used in the study, in both the original language and English, as Supporting Information, or include a citation if it has been published previously. A copy of interview guide has been added and uploaded as additional file

---

## [Decision Letter · Decision Letter 1]

13 Jan 2022

PONE-D-21-12824R1To assess the determinants of family planning uptake among women of reproductive age in rural settings, Morogoro Region, Tanzania. Protocol for a cross-sectional studyPLOS ONE

Dear Dr. machange,

Thank you for submitting your manuscript to PLOS ONE. After careful consideration, we feel that it has merit but does not fully meet PLOS ONE’s publication criteria as it currently stands. Therefore, we invite you to submit a revised version of the manuscript that addresses the points raised during the review process.

ACADEMIC EDITOR: From the authors' response, it is unclear whether the Authors have addressed all the Reviewers' comments. One of the reviewer was not satisfied. The Authors should present a rebuttal letter with responses to all the issues raised and not only to some of them. The presentation of the manuscript completely highlighted did not help. It would have been much more useful to highlight only changes. All the Reviewers and Editorial issues need to be addressed.

We look forward to receiving your revised manuscript.

Kind regards,

Paola Viganò

Academic Editor

PLOS ONE

Additional Editor Comments:

From the authors' response, it is unclear whether the Authors have addressed all the Reviewers' comments. One of the reviewer was not satisfied. The Authors should present a rebuttal letter with responses to all the issues raised and not only to some of them. The presentation of the manuscript completely highlighted did not help. It would have been much more useful to highlight only changes. All the Reviewers and Editorial issues need to be addressed.

Reviewers' comments:

Reviewer's Responses to Questions

**Comments to the Author**

1. Does the manuscript provide a valid rationale for the proposed study, with clearly identified and justified research questions?

Reviewer #1: Yes

Reviewer #2: Yes

2. Is the protocol technically sound and planned in a manner that will lead to a meaningful outcome and allow testing the stated hypotheses?

Reviewer #1: Yes

Reviewer #2: Yes

3. Is the methodology feasible and described in sufficient detail to allow the work to be replicable?

Reviewer #1: Yes

Reviewer #2: Yes

4. Have the authors described where all data underlying the findings will be made available when the study is complete?

Reviewer #1: Yes

Reviewer #2: Yes

5. Is the manuscript presented in an intelligible fashion and written in standard English?

Reviewer #1: Yes

Reviewer #2: Yes

6. Review Comments to the Author

You may also provide optional suggestions and comments to authors that they might find helpful in planning their study.

Reviewer #1: In general, the author has made a good revision. I enjoyed reading the revised study protocol, and the comments what we give also addressed and well presented.

Reviewer #2: I understand that my comments have been shared with the authors. However, I do not think the authors have seen my previous comments as I do not see any changes in the manuscript. If authors have seen my comments, and if they do not agree with them, I would appreciate it if they state why they do not agree with my comments/suggestions in the next "author's response" in a "point-by-point" manner.

7. PLOS authors have the option to publish the peer review history of their article (what does this mean?). If published, this will include your full peer review and any attached files.

Reviewer #1: **Yes: **Tesfalem Tilahun Yemane

Reviewer #2: No

---

## [Author Response · Author response to Decision Letter 1]

28 Feb 2022

POINT-BY-POINT RESPONSES TO REVIEWERS’ AND EDITOR’S COMMENTS

We would like to take this opportunity to thank you all, editors and reviewers from PLOS ONE We have appreciated all the comments given in order to improve the manuscripts. We have thoroughly responded to all reviewers’ comments. The table below is a summary of responses for the reviewers.

 Comments to the authors Authors’ Response

 General Authors had revised the whole document for grammar check and clarity

1.0 Second Round Reviewer’s comments 

 Reviewer #1 

1.1 In general, the author has made a good revision. I enjoyed reading the revised study protocol, and the comments what we give also addressed and well presented. Noted, Thank you 

 Reviewer #2 

1.2 I understand that my comments have been shared with the authors. However, I do not think the authors have seen my previous comments as I do not see any changes in the manuscript. If authors have seen my comments, and if they do not agree with them, I would appreciate it if they state why they do not agree with my comments/suggestions in the next "author's response" in a "point-by-point" manner. The comments were received. Authors worked on all the comments and the responses are given below serial number 3 of this document 

The corresponding revisions are highlighted in respective sections of the document 

2.0 Editor’s comments 

2.1 Please include additional information regarding the survey or questionnaire used in the study and ensure that you have provided sufficient details that others could replicate the analyses. For instance, if you developed a questionnaire as part of this study and it is not under a copyright more restrictive than CC-BY, please include a copy, in both the original language and English, as Supporting Information. If the original language is written in non-Latin characters, for example Amharic, Chinese, or Korean, please use a file format that ensures these characters are visible. The questionnaire description has been revised. The questionnaire has been adapted from the Demographic Health Surveys questionnaire. We only selected the questions related to family planning for this particular study. Minor modification has been done. 

The questionnaire has been uploaded as additional file (Appendix 1) (Page 6)

2.2 Please state whether you validated the questionnaire. Please provide details regarding the validation group within the methods section. The questionnaire was not validated because it was extracted from a standard questionnaire that is normally used for country wide health and demographic survey. 

However minor modifications has been made to match with the current study 

2.3 Please include a copy of the interview guide to be used in the study, in both the original language and English, as Supporting Information, or include a citation if it has been published previously. A copy of interview guide has been added and uploaded as additional file (Page 7)

3.0 First Reviewer’s comments 

 Reviewer #1 

3.1 The discussion section must be added.

This section should include the any issues involved in performing the study that are not covered in other sections. These can include: limitations of the study design & dissemination plans • The discussion part has been added (Page 10 - 12)

• Limitation of the study design and dissemination plans added page 12

• Dissemination plan has been added on page 9

 Reviewer # 2 

3.2 ABSTRACT:

Methodology

Authors indicate Objectives 1 through 3. However, readers do not know which objective you refer to 1, 2, or 3. Adding these numbers in the "Objective of the study" is recommended. Also, there is only one objective in the main text. Authors may consider including three specific objectives in the main text. The section has been revised accordingly to accommodate the comments.

The broad objective has been maintained at the abstract section Page 2 and at the end of the introduction section page 4

The specific objective have been stated in numbers on page 4 

3.3 Keywords: "Women of Reproductive Age" would be more suitable than "Reproductive age." The keyword has been revised. Page 2 and the word has been revised throughout the document

3.4 Citation: Please be thorough on the citation format. Sometimes authors use (author, reference number), which is not consistent. The citation format has been revised. Numbering style (Vancouver style) has been used throughout the document 

3.5 Introduction:

Women of Reproductive Age (WRA) appears first on this page. Please add an acronym where it first appears and be consistent in using it in subsequent text.. • We have agreed to maintain the long form of the term women of reproductive age because the abbreviation is not standard and we are trying to reduce the abbreviations 

 Authors define WRA are those aged 18-49. WHO and other UN agencies use the definition of WRA (15-49 years)? Please provide annotation or explanation • We have revised the section and agreed to use the standard definition of Women of reproduction age as per WHO, UN i.e 15 – 49 years Page 5

3.6 TDHS: where it first appears on page 4; please spell it out. It has been revised and the acronym has been written in long from first before the acronym Page 1

3.7 CycleBeads® is one word. This word has been removed 

3.8 Literature review:

 It is suggested that authors add a review of more literature on contraceptive use among WRA in Tanzania (esp. Morogoro or neighboring regions) as many studies have already been conducted. By doing this, the authors will have a much more vigorous justification for conducting this study (including highlighting why this study has to be conducted among WRA in Morogoro and those residing in rural areas). The literature review has been added in the introduction section. – Introduction section page 2 - 4

3.9 METHODS

Study Area: 

More description on Morogoro Region (the number of Districts within the Region, among which how many Districts are considered "rural," and how authors define "rural" in this study. Revised accordingly pages 3 – 4

The definition of rural is based on the current classification of these districts as per Report of Basic Demographic and Socio-Economic Profile of 2014

3.10 Research design: Do authors consider this study as a mixed-methods study? If so, please describe which design of a mixed-methods study the authors employ with its justification. Revised accordingly pages 5. This study will be cross-sectional with mixed approaches namely qualitative and quantitative 

3.11 Sampling Procedure for Qualitative Approach

Please describe in more detail which approach of purposive sampling the authors adopt for this study.

How do authors determine key informants from the results of the quantitative study?

Data Collection Methods for Quantitative Approach Described accordingly Methods section pages 5 - 6

3.12 What kind of questionnaire do the authors plan to use for this study? Has it been validated? The questionnaire has been adapted from the Demographic Health Surveys questionnaire. We only selected the questions related to family planning for this particular study. Minor modification has been done. 

The questionnaire has been uploaded as additional file (Appendix 1) (Page 6)

3.13 Data Collection Methods for Qualitative Data

No details are known, but the authors' interview method in this manuscript would be key informant interviews rather than in-depth interviews.

Please also indicate whether recording and transcriptions are made, how long does each interview would take, and if the authors plan to interview the informants only once or iteratively. Revised on page 7

3.14 Definition and Measurement of Variables

Would the dependent variables any FP use, rather than modern methods use? Definition of variables are provided on page 8

3.15 During the qualitative interviews, authors should highlight on positive aspects of FP use rather than just focusing on the challenges. The qualitative part is meant to explore both positives and negative aspects for broad understanding. Page 7

---

## [Decision Letter · Decision Letter 2]

1 Apr 2022

To assess the determinants of family planning uptake among women of reproductive age in rural settings, Morogoro Region, Tanzania. Protocol for a cross-sectional study

PONE-D-21-12824R2

Dear Dr. Machange,

We’re pleased to inform you that your manuscript has been judged scientifically suitable for publication and will be formally accepted for publication once it meets all outstanding technical requirements.

Kind regards,

Paola Viganò

Academic Editor

PLOS ONE

Additional Editor Comments (optional):

Reviewers' comments:

Reviewer's Responses to Questions

**Comments to the Author**

1. Does the manuscript provide a valid rationale for the proposed study, with clearly identified and justified research questions?

Reviewer #2: Yes

2. Is the protocol technically sound and planned in a manner that will lead to a meaningful outcome and allow testing the stated hypotheses?

Reviewer #2: Yes

3. Is the methodology feasible and described in sufficient detail to allow the work to be replicable?

Reviewer #2: Yes

4. Have the authors described where all data underlying the findings will be made available when the study is complete?

Reviewer #2: Yes

5. Is the manuscript presented in an intelligible fashion and written in standard English?

Reviewer #2: Yes

6. Review Comments to the Author

You may also provide optional suggestions and comments to authors that they might find helpful in planning their study.

Reviewer #2: Thank you very much for responding to the previous questions in a point-by-point manner, which is well reflected in the improved quality of the manuscript. The reviewer wishes the best to the authors for implementing this study.

7. PLOS authors have the option to publish the peer review history of their article (what does this mean?). If published, this will include your full peer review and any attached files.

Reviewer #2: No

---

## [Editor Report · Acceptance letter]

7 Apr 2022

PONE-D-21-12824R2 

To assess the determinants of family planning uptake among women of reproductive age in rural settings, Morogoro Region, Tanzania. Protocol for a cross-sectional study

Dear Dr. Machange:

I'm pleased to inform you that your manuscript has been deemed suitable for publication in PLOS ONE. Congratulations! Your manuscript is now with our production department. 

Kind regards, 

on behalf of

Dr. Paola Viganò 

Academic Editor

PLOS ONE